# The Impact of the FIFA 11+ Neuromuscular Training Programme on Ankle Injury Reduction in Football Players: A Systematic Review and Meta-Analysis

**DOI:** 10.3390/muscles4030030

**Published:** 2025-08-12

**Authors:** Can Eser, Türker Bıyıklı, Paul J. Byrne, John D. Duggan, Joseph I. Esformes, Jeremy A. Moody

**Affiliations:** 1Cardiff School of Sport and Health Sciences, Cardiff Metropolitan University, Cardiff CF5 2YB, UK; can.eser@outlook.com.tr (C.E.); jesformes@cardiffmet.ac.uk (J.I.E.); 2Department of Coaching Education, Faculty of Sports Sciences, Marmara University, 34815 Istanbul, Türkiye; turker.biyikli@marmara.edu.tr; 3Department of Health and Sport Sciences, South East Technological University, R93 V960 Carlow, Ireland; 4Department of Sport, Exercise & Nutrition, School of Science & Computing, Atlantic Technological University, H91 T8NW Galway, Ireland; john.duggan@atu.ie; 5School of Physical Education and Sports, Nisantasi University, 34398 Istanbul, Türkiye

**Keywords:** football injuries, ankle injuries, injury prevention, injury incidence, neuromuscular training, athletic performance, muscle strength, proprioception, plyometrics, muscular function

## Abstract

This systematic review and meta-analysis was conducted to evaluate the effectiveness of the FIFA 11+ injury prevention programme, a neuromuscular training intervention involving muscular strength, proprioception, balance, and plyometric exercises, in reducing ankle injury incidence among football players. Included are randomised controlled trials (RCTs) involving players of any age, sex, or competition level, comparing the FIFA 11+ programme with standard warm-up routines. Studies were eligible if they had a minimum follow-up of five months and reported at least two of the following: number of ankle injuries, incidence rate, and exposure hours. Searches were conducted in PubMed, MEDLINE (via OVID), Scopus, and SPORTDiscus up to October 2024. Risk of bias was assessed using the revised Cochrane Risk of Bias tool (RoB 2), and a random-effects meta-analysis was conducted. Three RCTs, involving 3833 participants and 286,827 exposure hours, met the inclusion criteria. The pooled analysis showed that the FIFA 11+ programme significantly reduced ankle injury rates compared to control groups (rate ratio (RR) = 0.67, 95% confidence interval (CI): 0.46–0.96, *p* = 0.03, I^2^ = 52%), reflecting a 33% lower risk. The certainty of the evidence was rated as moderate due to bias in two studies. In conclusion, the FIFA 11+ programme significantly reduces ankle injury incidence and supports implementation at all levels. However, further research is needed to examine long-term effects, optimal doses, and applicability across diverse football populations.

## 1. Introduction

Sports participation is widely recognised for its numerous health, social, and economic benefits; however, it also comes with an inherent risk of injury [1]. Sports injuries not only affect individual athletes’ health and well-being but also impose significant financial burdens on teams and healthcare systems globally [2,3]. Among sports, football (soccer) is the most popular sport globally and contributes approximately USD 250 billion annually to the global sports industry’s revenue [4]. As reported by FIFA, over 270 million individuals worldwide, approximately 4% of the global population, are engaged in football in some capacity, including as players, fans, or support staff [5,6]. The popularity and financial significance of football underscore the necessity of maintaining player health, particularly at the elite level, where success is closely linked to player availability [7].

In football, the overall injury rate is estimated as 8.1 per 1000 exposure hours, with match play showing nearly a tenfold increase compared to training [8]. These injuries predominantly occur in the lower extremities, with an overall incidence of 6.8 per 1000 h [8]. Among lower limb injuries, the thigh is the most commonly affected region, with an incidence rate of 1.8 injuries per 1000 h of exposure, followed by the knee at 1.2 injuries and the ankle at 1.1 injuries [8]. Muscle and tendon injuries are the most frequent type, with an incidence rate of 4.6 per 1000 h. Contusions follow with a rate of 1.4, and joint or ligament injuries occur at a rate of 0.4 per 1000 h [8]. Most of these injuries are classified as mild to moderate, resulting in a time loss of fewer than four weeks [9].

Among the various injuries seen in football, ankle injuries are a particular concern due to their high incidence and potential for recurrence [8,10]. Sprains are the most common type of ankle injury, accounting for 68% of all cases, with lateral ligament sprains representing the predominant subtype [10]. These injuries often affect the dominant leg, with approximately 40% associated with foul play involving forced ankle inversion [10,11]. The time needed to return to play varies depending on the specific diagnosis. Players’ absence following lateral ligament sprains lasts around two weeks, whereas high ankle sprains average more than six weeks [9,10]. These findings emphasise the need for effective preventative strategies for ankle injuries.

Several injury prevention programmes have been developed to address this issue [12]. The most common elements of these programmes include proprioceptive training, balance exercises (such as wobble boards and single-leg stands), and strengthening exercises targeting the muscles surrounding the ankle [12,13]. Furthermore, dynamic stretching and mobility exercises are frequently incorporated to enhance flexibility and range of motion, thereby reducing the risk of injuries [12,14].

One widely recognised injury prevention programme is the FIFA 11+, which comprises three distinct components: dynamic-based movements; core, strength, plyometrics, and balance exercises with three levels of progression; and three higher-intensity running mechanics drills (Table 1) [15,16]. Once players become familiar with the drills, the entire warm-up routine takes approximately 20 min, and exercises are typically performed in pairs, with no additional equipment required [15]. The effectiveness of the FIFA 11+ programme has been validated across various populations and levels of play [17]. For instance, Soligard et al. [16] conducted a cluster-randomised trial involving 1892 female youth footballers and found a 32% reduction in the overall risk of injury among those assigned to the FIFA 11+ warm-up group. Similarly, Silvers-Granelli et al. [18] demonstrated a 46.1% reduction in overall injury incidence among male collegiate football players who used the FIFA 11+ programme, including a 35% reduction in ankle injuries. Furthermore, Steffen et al. [19] found that high adherence to the FIFA 11+ programme was associated with a significantly lower injury risk, underscoring the importance of consistent implementation. Collectively, these findings support the widespread use of the FIFA 11+ as an accessible, low-cost, and evidence-based injury prevention programme [16,17,18,19].

Despite numerous systematic reviews examining the overall impact of the FIFA 11+ programme on injury prevention, ankle injuries have not yet been specifically evaluated, despite their high incidence and recurrence rate [17,20]. Given the substantial impact ankle injuries can have on player availability, team performance, and economic burdens for football teams and organisations, a dedicated analysis of the effectiveness of the FIFA 11+ programme in reducing ankle injuries is critical [2,7,9,10]. This systematic review and meta-analysis aims to fill this research gap by synthesising evidence from randomised controlled trials to determine whether the FIFA 11+ effectively reduces ankle injury incidence compared to standard warm-up routines. Clarifying this relationship could guide coaches, sports scientists, and clinicians in adopting targeted prevention strategies, ultimately enhancing player health, reducing injury-associated costs, and improving competitive performance [2,7,17].

## 2. Methods

This systematic review was conducted following the PRISMA (Preferred Reporting Items for Systematic Reviews and Meta-Analyses) 2020 statement [21].

### 2.1. Eligibility Criteria

The eligibility criteria for this systematic review were defined using the PICO framework, as presented in Table 2 [22]. The studies included involved football players at any age, of any sex, and at any level of competition. The synthesis was based on comparisons of intervention groups that used the FIFA 11+ programme versus control groups that used usual or standard warm-up routines. Modifications to the FIFA 11+ programme were not included in the analysis. To be included in the review, studies had to report at least two of the following outcomes: the number of ankle injuries, the incidence rate of ankle injuries, and the number of hours of exposure. Ankle injuries were defined as those that resulted in players being unable to participate in the subsequent training session or match [23]. Exposure hours were defined as the total time spent by players in training or at competitions throughout the study period [16].

Eligible study designs included randomised controlled trials (including those with cluster designs) with a minimum follow-up period of five months (half a season). The review included studies published in the English language from 2006 onwards, the year the FIFA 11+ programme was launched. Manuscripts, abstracts, and reports that were not yet published, as well as those that were not peer-reviewed, were excluded. Studies that did not measure or report the primary outcomes of interest were considered ineligible.

### 2.2. Information Sources

The most recent date on which all sources for this systematic review were searched was 2 October 2024. The bibliographic databases utilised in this systematic review were PubMed, MEDLINE (via OVID), Scopus, and SPORTDiscus, with the full range of available data in each platform. In addition, the reference lists of the eligible articles were reviewed to identify additional relevant studies.

### 2.3. Search Strategy

The comprehensive search strategy for each database is presented in Table 3. In PubMed, the search strategy included terms such as “football”, “soccer”, “F-MARC”, “FIFA 11+”, “warm-up programme”, “injury prevention programme”, “injury”, “injuries”, “randomised controlled trial”, and their synonyms. These terms were combined with a set of Medical Subject Headings (MeSH) terms and free-text terms. The search terms were similarly adapted for MEDLINE (via OVID), Scopus, and SPORTDiscus, using the syntax specific to each platform. In Scopus, the search was restricted to the title, abstract, and keyword fields, whereas in other databases, searches were conducted across all fields. To ensure the integrity of the review process, eligibility criteria were applied to the search restrictions. Only studies published in the English language from 2006 onwards were included.

Systematic reviews and meta-analyses on similar topics were consulted to validate the search strategy, and a small number of studies that met the eligibility criteria were identified. These studies were then cross-checked to ensure they could be retrieved using the search terms and strategy developed for this review.

### 2.4. Selection Process

The screening process for this systematic review was managed using Endnote21 software (Version 21.4, Clarivate) to handle references, remove duplicates, and organise studies. Two reviewers (CE and JIE) independently screened the titles and abstracts of the retrieved articles for eligibility based on the inclusion criteria. Full-text screening was also performed independently by CE and JIE, and disagreements were resolved through discussion with a third reviewer (JAM).

### 2.5. Data Collection Process

Data extraction was conducted independently by two reviewers using a standardised data extraction form created in Excel. In instances in which uncertainty arose regarding the interpretation or completeness of the data, the reviewers consulted the senior author to resolve any discrepancies and reach a final decision.

### 2.6. Data Items

The primary metrics sought in this review were the number of ankle injuries, the incidence rate of ankle injuries, and the number of hours of exposure. These outcomes were selected as they serve as critical indicators of the efficacy of the FIFA 11+ injury prevention programme in reducing ankle injury incidence among football players.

The inclusion and definition of outcome domains remained unchanged throughout the review, and all planned initial outcome measures were retained. Similarly, the method to select results within the eligible outcome domains remained consistent throughout the review. Additionally, data were collected on additional variables, including participant characteristics (e.g., age, gender, and level of competition) and intervention specifics (e.g., duration and frequency of FIFA 11+ sessions).

### 2.7. Study Risk of Bias Assessment

A revised Cochrane Risk of Bias tool (RoB 2, 2019) was used to assess the risk of bias in each included study [24]. The RoB 2 assesses five domains: the randomisation process, deviations from the intended interventions, missing outcome data, measurement of the outcome, and selection of the reported result. Each domain was assessed and classified as exhibiting a low risk of bias, some concerns, or a high risk of bias.

An overall risk of bias judgement was made by assigning the highest level of bias observed in any domain to the overall risk of bias of the study, which was classified as “low risk”, “some concerns”, or “high risk”. The risk of bias was evaluated by two independent reviewers. In instances of ambiguity, the reviewers consulted the senior author to resolve discrepancies and reach a final decision.

### 2.8. Effect Measures

The primary effect measure was the RR to analyse ankle injury incidence per 1000 exposure hours. The rate ratio was calculated by comparing the incidence of ankle injuries between the intervention group (FIFA 11+ programme) and the control group (standard warm-up). This allowed for comparing the relative reduction in ankle injury incidence associated with the FIFA 11+ programme. The meta-analysis synthesised the RRs across studies, and 95% confidence intervals (CIs) were employed to indicate the precision of the estimates. An RR of 1 indicates no difference in injury rates between the groups, whereas values less than 1 suggest a reduced injury rate, and values greater than 1 indicate an increased injury rate [25].

### 2.9. Synthesis Methods

The findings of each study are presented in a systematic tabular format. This included the number of ankle injuries, total exposure hours, and ankle injury rates per 1000 exposure hours in both the FIFA 11+ and control groups. Furthermore, RRs and their 95% CIs are presented in the same table for each study.

A random-effects meta-analysis was conducted to synthesise the data quantitatively, utilising the Comprehensive Meta-Analysis software (Version 4, Biostat). The meta-analysis employed the number of ankle injuries and total exposure hours in both the intervention and control groups to calculate RRs. The I^2^ statistic was used to quantify statistical heterogeneity. In accordance with the Cochrane Handbook, values of 0–40% might not be important; 30–60% may represent moderate heterogeneity; 50–90% may represent substantial heterogeneity; and 75–100% may represent considerable heterogeneity, depending on context [26]. Sensitivity analysis was conducted by excluding studies at high risk of bias. The meta-analysis results are illustrated in forest plots, displaying the individual study RRs and the overall pooled effect size, with studies arranged chronologically by year of publication.

### 2.10. Reporting Bias Assessment

Given the small number of included studies, a formal statistical assessment of reporting bias (such as Egger’s test or visual inspection of funnel plots) was not conducted, following recommendations from the Cochrane Handbook [27]. Additionally, selective outcome reporting assessment by comparing study protocols or published methods to reported outcomes was not feasible due to the unavailability of study protocols [27].

### 2.11. Certainty Assessment

The certainty of the evidence in this systematic review was evaluated using the Grading of Recommendations Assessment, Development, and Evaluation (GRADE) system (2019 version) [28,29]. This tool assesses several factors that can affect the overall certainty of the evidence, including the precision of the effect estimate, the consistency of the findings across studies, the directness of the evidence, and the risk of bias. Each domain was carefully considered, and studies were evaluated based on their performance across these factors.

The decision rules for arriving at an overall rating were based on the GRADE approach, with evidence classified as having high, moderate, low, or very low certainty [28]. Two independent reviewers conducted the assessment. In cases of uncertainty, the reviewers consulted the senior author to resolve any discrepancies and reach a final decision. The results of these assessments were expressed using standardised GRADE phraseology, such as “probably reduces risk”, to reflect the level of certainty and to ensure consistent interpretation across outcomes [28].

## 3. Results

### 3.1. Study Selection

A total of 488 records were identified through the database searches. After removing 273 duplicates, the remaining 215 records were subjected to title and abstract screening. During the screening process, 203 records were excluded based on irrelevance to the review criteria. Subsequently, the eligibility of 12 full-text articles was assessed, resulting in the exclusion of seven studies due to intervention-related reasons, two studies due to the absence of ankle injury data, and one study due to issues related to study design. In conclusion, two studies met the inclusion criteria, with an additional study identified through reference lists, resulting in three studies included in the review. The three studies were also included in the meta-analysis. Figure 1 presents a PRISMA flow diagram that outlines the study selection process [21].

### 3.2. Study Characteristics

Table 4 provides an overview of each study, including details on the study design, participant characteristics, specifics of the intervention, and outcome measures. All studies employed the standard FIFA 11+ programme and compared it with typical or unstructured warm-up routines in the control groups. Furthermore, the table provides an overview of the intervention details, including the frequency and follow-up periods. The frequency of the intervention varied from two to five times per week, while the follow-up periods ranged from five to eight months.

### 3.3. Risk of Bias in Studies

The risk of bias for the included studies is presented in Figure 2. All three studies exhibited a low risk of bias about the domains of the randomisation process and the selection of reported outcomes. The randomisation procedures were conducted appropriately, and the pre-specified protocols reported the outcome measures. Regarding deviations from the intended interventions, all studies were assessed as having a low risk of bias due to high adherence to the intervention protocols. However, the study by Silvers-Granelli et al. [18] was evaluated as having a high risk of bias in the domain of missing outcome data. This was primarily due to the dropout rate and incomplete follow-up, which likely affected the study’s results.

In contrast, Soligard et al. [16] and Owoeye et al. [30] were assessed as having a low risk of missing data. Regarding outcome measurement, Owoeye et al. [30] and Silvers-Granelli et al. [18] were rated with some concerns. The lack of blinding of outcome assessors in these studies introduced the potential for bias, as unblinded assessors may have influenced how injuries were reported. Conversely, Soligard et al. [16] was evaluated as having a low risk of bias in this domain due to the implementation of standardised injury reporting with blinded assessors.

### 3.4. Results of Individual Studies

Silvers-Granelli et al. [18] reported 59 ankle injuries out of 675 participants in the intervention group and 115 out of 850 participants in the control group, corresponding to 1.675 and 2.601 injuries per 1000 h of exposure, respectively. The effect estimate for this study was an incidence rate ratio (IRR) of 0.65 (95% CI: 0.48–0.87). Similarly, Owoeye et al. [30] reported 10 ankle injuries in the intervention group (*n* = 212) during 51,017 exposure hours (0.196 injuries per 1000 exposure hours) and 30 injuries in the control group (*n* = 204) during 61,045 exposure hours (0.491 injuries per 1000 exposure hours), with an IRR of 0.40 (95% CI: 0.19–0.82). Soligard et al. [16] found 51 ankle injuries in the intervention group (*n* = 1055) and 52 in the control group (*n* = 837), with incidence rates of 1.022 and 1.145 per 1000 exposure hours, respectively, yielding an IRR of 0.89 (95% CI: 0.61–1.31). These results are presented in Table 5.

### 3.5. Results of Synthesis

A random-effects meta-analysis was conducted using the RR as the effect measure based on data from three studies that evaluated the impact of the FIFA 11+ programme on ankle injury rates per 1000 exposure hours. The pooled RR was 0.67 (95% CI: 0.46–0.96, *p* = 0.03, I^2^ = 52%), with moderate heterogeneity, indicating a 33% reduction in ankle injury rates in the FIFA 11+ group compared to the control group (Figure 3). The analysis included a total of 286,827 exposure hours across the three studies.

A sensitivity analysis was conducted, excluding studies with a high risk of bias. This resulted in a pooled RR of 0.63 (95% CI: 0.29–1.38, *p* = 0.25, I^2^ = 73%), with higher heterogeneity (Figure 4). These findings suggest that differences in study populations or intervention adherence may have influenced the variability in effect sizes. While the sensitivity analysis indicated a non-significant effect, it still showed a trend toward reduced ankle injury rates with the FIFA 11+ programme.

### 3.6. Reporting Biases

Due to the inclusion of only three studies, a formal statistical assessment of reporting bias (e.g., Egger’s test or funnel plot analysis) was not feasible, in line with the guidance in the Cochrane Handbook [27]. Additionally, selective outcome reporting could not be evaluated due to the lack of publicly accessible protocols. Although Soligard et al. [16] was registered (ISRCTN10306290), a complete protocol or statistical analysis plan was not available. Silvers-Granelli et al. [18] and Owoeye et al. [30] did not report trial registration or provide protocol information. Consequently, the risk of reporting bias across the synthesis remains unclear.

### 3.7. Certainty of Evidence

The overall certainty of the evidence for the primary outcome, the number of ankle injuries, was assessed using the GRADE approach. The certainty was rated as moderate, mainly due to concerns about the risk of bias in the included studies. The reasons for downgrading the level of certainty were explained in footnotes within the Summary of Findings (SoF) table (Table 6), with the lack of blinding in outcome assessments and missing data being the main factors reducing the level of certainty.

**Risk of Bias:** The risk of bias was downgraded by one level due to concerns regarding two of the included studies (Silvers-Granelli et al. [18] and Owoeye et al. [30]) for the lack of blinding in outcome assessments and missing outcome data. These limitations may have affected the reliability of the reported results.

**Consistency:** Despite moderate heterogeneity (I^2^ = 52%), downgrading for inconsistency was not warranted. The direction of the effect was consistent across all studies, with the FIFA 11+ programme demonstrating a favourable outcome. The observed variability in effect sizes was attributable to differences in population characteristics.

**Imprecision:** Downgrading for imprecision was not warranted. The CIs in the primary meta-analysis (RR = 0.67, 95% CI: 0.46–0.96) did not cross the line of no effect, indicating that the estimate of the intervention effect was sufficiently precise.

**Indirectness:** Downgrading for indirectness was not warranted. All studies addressed the research question, focusing on football players, the FIFA 11+ intervention, and ankle injury outcomes.

**Publication bias:** Due to the small number of studies included (*n* = 3), tests like funnel plots were unreliable for detecting publication bias and thus were not assessed.

## 4. Discussion

This systematic review, to the authors’ knowledge, is the first to evaluate specifically the impact of the standard FIFA 11+ programme on the incidence of ankle injuries in football players. Previous reviews, such as Al Attar et al. [12], also examined the FIFA 11+ programme but included modified versions, such as the FIFA 11+ Kids and a shortened 10-min post-training version. This meta-analysis, which focused exclusively on the standard FIFA 11+ programme, showed a 33% reduction in the incidence of ankle injuries, with a pooled RR of 0.67 (95% CI: 0.46–0.96). This finding is consistent with that of Al Attar et al. [12], which showed a 36% reduction in ankle injuries per 1000 exposure hours (IRR = 0.64, 95% CI: 0.54–0.77) in programmes that included balance exercises such as the single-leg stance, one of the exercises in the FIFA 11+ programme.

Similarly, Sadigursky et al. [17] reinforce the present findings, reporting a 30% reduction in all injuries (RR = 0.70, 95% CI: 0.52–0.93, *p* = 0.01) among 6344 football players. This provides additional support for the broad applicability of the FIFA 11+ programme across different populations and competition levels. Moreover, Al Attar et al. [31] explored an additional layer of application by comparing the standard programme with a modified post-training version. In this study, both the control and experimental groups completed the standard FIFA 11+ programme before training. However, after training, the experimental group also completed an additional 10 min of the modified FIFA 11+ programme. Their results showed that the experimental group, which performed both pre- and post-training exercises, had a significantly lower injury incidence (0.081 injuries per 1000 exposure hours) than the control group, which only performed the FIFA 11+ pre-training programme (0.324 injuries per 1000 exposure hours). Similarly, Steffen et al. [19] found that players who adhered consistently to the programme had a markedly lower overall injury risk (RR = 0.28, 95% CI: 0.10–0.79). Although these studies were excluded from our review due to eligibility criteria, they raise important considerations about whether greater adherence or cumulative exposure to neuromuscular training, achieved through an increased frequency or a higher exercise volume, may optimise injury reduction outcomes.

Ankle sprains are the most common type of ankle injury in football players, accounting for 68% of all ankle injuries [10]. Deficits in dynamic balance, often measured by the Y-Balance Test or Star Excursion Balance Test (SEBT), are considered to be one of the contributing risk factors for these injuries. Mason et al. [32] showed that reduced anterior and posterior-lateral reach distances in male athletes were significantly associated with a higher risk of ankle sprains, suggesting that poor dynamic balance plays a role in ankle injury susceptibility. Steffen et al. [19] found that players with high adherence to the FIFA 11+ programme significantly improved dynamic balance, with better results in five out of six SEBT directions. These findings underline the capacity of the FIFA 11+ programme to address known biomechanical deficits that predispose players to ankle injuries.

Although the meta-analysis indicates a statistically significant reduction in ankle injury incidence, several limitations warrant a cautious interpretation of the findings. This meta-analysis was based on only three RCTs, which limits the statistical power and generalisability of the findings. While the pooled estimate reached statistical significance, the precision of the effect remains constrained by the limited study pool. Furthermore, as outlined in the Cochrane Handbook, formal assessments of publication bias, such as funnel plots or Egger’s test, are not recommended when fewer than ten studies are available, due to their low reliability [27]. Although not formally assessed, the possibility of publication bias cannot be excluded. In the future, systematic reviews might consider including high-quality non-randomised studies (e.g., prospective cohorts), particularly in elite or professional football settings in which RCTs are difficult to implement due to ethical and logistical constraints.

Two of the three included studies, Silvers-Granelli et al. [18] and Owoeye et al. [30], exhibited notable methodological weaknesses, particularly relating to missing outcome data and the lack of blinding in outcome assessments. These issues may have introduced bias and potentially inflated the observed protective effect. The moderate heterogeneity observed in the meta-analysis (I^2^ = 52%) may be explained by both clinical and methodological differences across the included studies. Participant characteristics such as age, sex, body mass index, and playing level varied considerably, ranging from adolescent female players in organised youth leagues [16] to collegiate male athletes in the United States [18] and male youth players in a Nigerian junior league [30]. These differences are known to influence injury risk and responsiveness to preventive interventions [32,33].

The interventions’ characteristics also differed, with the FIFA 11+ programme implemented for periods ranging from five to eight months, and with the session frequency ranging from two to five times per week [16,18,30]. While all studies followed the standard FIFA 11+ protocol, this variation in frequency and duration may have influenced the magnitude of the observed effect. However, none of the studies conducted stratified analyses based on adherence level, limiting our understanding of potential dose–response relationships. Adherence contributes to the overall “dose” of neuromuscular training received, which may affect the intervention’s protective benefit. Yet, the absence of stratified data prevents firm conclusions about whether increased exposure leads to proportionally greater injury reduction. Additionally, none of the included studies conducted long-term follow-up beyond the immediate post-intervention period, leaving the durability of the FIFA 11+ programme’s protective effects uncertain. The observed short-term reductions are promising; however, whether these benefits persist once players discontinue structured neuromuscular training remains an open question.

From a practical perspective, these findings suggest that coaches should prioritise the consistent and structured delivery of the FIFA 11+ programme to maximise its effectiveness. Implementation at least twice per week is recommended, as this was the minimum frequency reported in studies demonstrating benefit [16,18,30]. Although no study specifically analysed the relationship between adherence and ankle injury outcomes, the association between high adherence and improved dynamic balance observed by Steffen et al. [19] reinforces the value of regular implementation. Coaches should therefore prioritise exercises that target balance, proprioception, and neuromuscular control, as these components directly address deficits such as joint instability that are commonly associated with ankle sprain risk [12,19,32].

Beyond the limitations of the included studies, broader issues within the injury prevention literature should be acknowledged. Many injury prevention programmes, including the FIFA 11+, demonstrate strong efficacy under controlled trial conditions, yet face significant barriers when it comes to their real-world implementation [34]. Variability in coaching practices, player motivation, organisational support, and cultural attitudes towards warm-up routines can undermine adherence and effectiveness outside the research setting [34]. Furthermore, the majority of available evidence derives from youth and amateur populations, with limited high-quality research focused on elite and professional footballers, who may have different risk profiles, exposure levels, and physiological demands [35]. This scarcity of elite-focused research constrains the generalisability of current injury prevention recommendations.

Additionally, outcome reporting remains inconsistent across studies, with differences in injury definitions, exposure calculations, and follow-up durations impeding meta-analytical comparisons and weakening the robustness of evidence synthesis [16,18,30]. To address these inconsistencies, greater adoption of standardised injury definitions and reporting guidelines is necessary. The FIFA Consensus Statements, such as the 2006 “Consensus Statement on Injury Definitions and Data Collection Procedures in Football” [23], provide comprehensive frameworks to harmonise injury surveillance methodologies. By applying such standardised criteria consistently across future trials, comparability between studies could be improved, facilitating more robust meta-analytical syntheses and enhancing the overall quality of the evidence base.

Another gap in the literature concerns the mechanisms underlying ankle injuries. Few studies have differentiated between contact and non-contact sprains, and recurrence rates have not been explored in sufficient depth [32,33]. A better understanding of injury mechanisms could inform more targeted intervention strategies.

While this review focused on the incidence of ankle injuries, it is important to recognise that injury severity and burden also play a critical role in understanding the overall impact of injury prevention programmes. As Verhagen et al. [36] argue, relying solely on incidence data may lead to an incomplete or even misleading interpretation of an intervention’s effectiveness. They emphasise that severity and burden, which incorporate factors such as time-loss and functional limitations, provide a more comprehensive picture of injury impact. These measures are closely related to availability for training and competition and may therefore offer more practical relevance for coaching and clinical decision making. Future studies should aim to report incidence, severity, and burden collectively to better evaluate the real-world effectiveness of injury prevention strategies.

Despite these limitations, the findings of this review strongly support the incorporation of the FIFA 11+ programme as a standard component of football training routines across all levels of competition. Coaches, sports scientists, and medical staff are encouraged to implement the FIFA 11+ consistently to mitigate the risk of ankle injuries. Moreover, national and regional football organisations should actively promote the adoption of this evidence-based intervention, particularly among youth and amateur athletes, whose injury burdens are high and access to rehabilitation resources may be limited [34]. The successful integration of the programme could reduce ankle injury incidence and potentially diminish the longer-term health consequences associated with chronic ankle instability and recurrent sprains [37].

Looking forward, future research should prioritise large-scale, high-quality randomised controlled trials with extended follow-up periods to establish the long-term sustainability of the FIFA 11+ programme’s protective effects. Investigations should also explore the programme’s applicability across diverse football populations, including professionals and players with a history of prior ankle injuries. There is also a need to examine the optimal frequency and intensity of programme delivery to maximise injury prevention benefits. Finally, future studies should seek to clarify the specific contributions of individual programme components, which could enable the tailoring of interventions to different risk profiles and playing environments.

## 5. Conclusions

In conclusion, the results of this systematic review and meta-analysis demonstrate that the FIFA 11+ injury prevention programme is effective in reducing the incidence of ankle injuries among football players. The pooled RR indicates a 33% reduction in injury rates, which is a statistically significant finding. This highlights the potential of the programme to serve as a valuable tool for injury prevention across various levels of competition. While the overall certainty of evidence is moderate due to some methodological concerns, the results support the incorporation of the FIFA 11+ programme into regular football training sessions. Further research is required to explore the programme’s efficacy across different populations, including professional players and those with previous injuries, and to determine the optimal programme frequency.

Beyond injury prevention, the FIFA 11+ programme may also have value within return-to-play protocols following ankle injuries. Incorporating balance, strength, and proprioceptive exercises from the FIFA 11+ into rehabilitation strategies could assist in restoring functional movement patterns, reducing the risk of recurrence, and enhancing player availability and long-term performance outcomes.

## Figures and Tables

**Figure 1 muscles-04-00030-f001:**
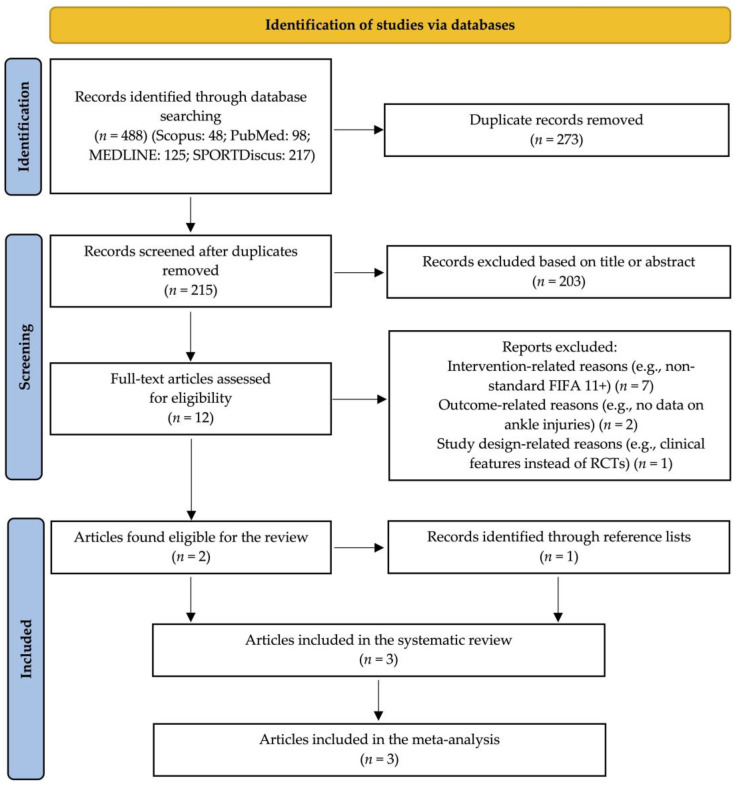
PRISMA flow diagram for the study selection process [21].

**Figure 2 muscles-04-00030-f002:**
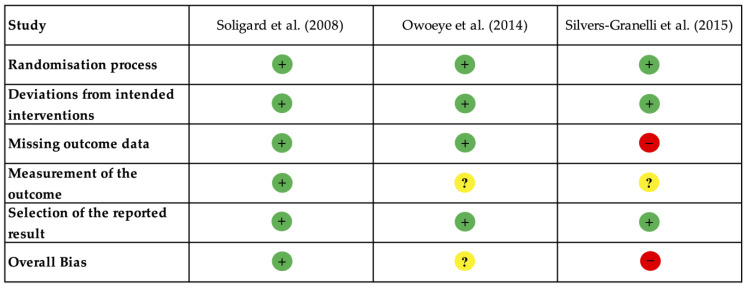
Risk of bias assessment for the included studies using the Revised Cochrane Risk of Bias Tool (RoB 2) [24]. Symbols indicate levels of bias: low risk (+), some concerns (?), and high risk (−). The included studies are Soligard et al. [16], Owoeye et al. [30], and Silvers-Granelli et al. [18].

**Figure 3 muscles-04-00030-f003:**
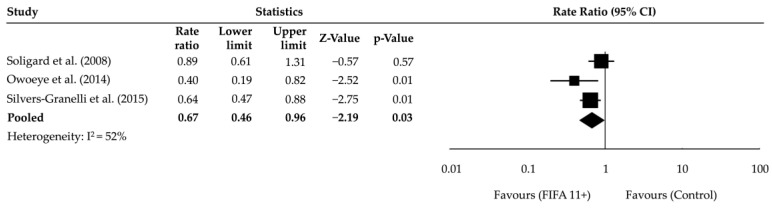
Forest plot of the meta-analysis showing RRs and 95% CIs for ankle injury incidence between intervention (FIFA 11+) and control groups. The pooled effect favours the FIFA 11+ programme. The included studies are Soligard et al. [16], Owoeye et al. [30], and Silvers-Granelli et al. [18].

**Figure 4 muscles-04-00030-f004:**
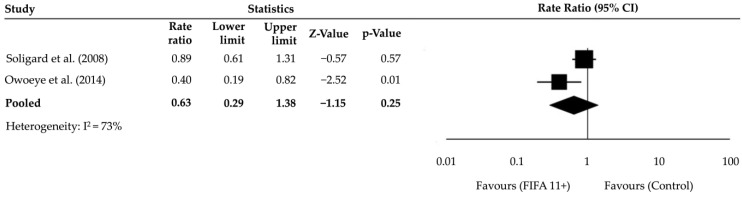
The sensitivity analysis forest plot, excluding studies at high risk of bias, presents the pooled RRs along with their 95% CIs. The included studies are Soligard et al. [16] and Owoeye et al. [30].

**Table 1 muscles-04-00030-t001:** Exercises, repetitions, and duration of the FIFA 11+ injury prevention programme [15,16].

Exercise Type	Repetitions/Duration
**Part 1—Running Exercises—8 Min**	
1. Running—Straight Ahead	2 sets
2. Running—Hip Out	2 sets
3. Running—Hip In	2 sets
4. Running—Circling Partner	2 sets
5. Running—Shoulder Contact	2 sets
6. Running—Quick Forwards and Backwards	2 sets
**Part 2—Core + Strength + Plyometrics + Balance—10 Min**	
7. The Bench	
Level 1: Static	3 × 20–30 s
Level 2: Alternate Legs	3 × 40–60 s
Level 3: One Leg Lift and Hold	3 × 20–30 s (for each leg)
8. Sideways Bench	
Level 1: Static	3 × 20–30 s (for each side)
Level 2: Raise and Lower Hip	3 × 20–30 s (for each side)
Level 3: With Leg Lift	3 × 20–30 s (for each side)
9. Hamstrings	
Level 1: Beginner	1 × 3–5 reps or 60 s
Level 2: Intermediate	1 × 7–10 reps or 60 s
Level 3: Advanced	1 × 12–15 reps or 60 s
10. Single Leg Stance	
Level 1: Hold The Ball	2 × 30 s (for each leg)
Level 2: Throwing Ball with Partner	2 × 30 s (for each leg)
Level 3: Test Your Partner	2 × 30 s (for each leg)
11. Squats	
Level 1: With Toe Raise	2 × 30 s
Level 2: Walking Lunges	2 × 10 reps (for each leg)
Level 3: One Leg Squats	2 × 10 reps (for each leg)
12. Jumping	
Level 1: Vertical Jumps	2 × 30 s
Level 2: Lateral Jumps	2 × 30 s
Level 3: Box Jumps	2 × 30 s
**Part 3—Running Exercises—2 Min**	
13. Running—Across the Pitch	2 sets
14. Running—Bounding	2 sets
15. Running—Plant and Cut	2 sets

**Table 2 muscles-04-00030-t002:** Selection criteria for the systematic review based on the PICO framework.

Criteria	Inclusion Criteria	Exclusion Criteria
Population	Football players at any age, of any sex, and at any competition level.	Studies not involving football players.
Intervention	FIFA 11+ injury prevention programme (standard version).	Modified versions of the FIFA 11+ programme.
Comparison	Usual or standard warm-up programmes (non-FIFA 11+).	Absence of control group.
Outcomes	Report at least two of the following outcomes: Number of ankle injuries; Incidence rate of ankle injuries; Exposure hours.	Studies that do not measure or report at least two of the specified outcomes.
Study Design	(Cluster) randomised controlled trials.	Studies with other designs.

**Table 3 muscles-04-00030-t003:** Search strategy for each database used in the systematic review.

Date of the Search	Databases	Keywords	Database Fields for the Search	Restrictions for the Search	Examples of Search Strategy Code Line
2 October 2024	PubMed	“football”, “soccer”, “F-MARC”, “FIFA 11+”, “The 11+”, “warm-up programme”, “injury prevention programme”, “injury”, “injuries”, “RCT”, “clinical trial”	All	Language: English Year: 2006 onwards	(“football” [MeSH Terms] OR “soccer” [MeSH Terms] OR “football” [All Fields] OR “soccer” [All Fields]) AND (“F-MARC” [All Fields] OR “FIFA 11+” [All Fields] OR “The 11+” [All Fields] OR “warm-up programme” [All Fields] OR “injury prevention programme” [All Fields]) AND (“injury” [All Fields] OR “injuries” [All Fields]) AND (“randomized controlled trial” [Publication Type] OR “RCT” [All Fields] OR “clinical trial” [Publication Type] OR “randomized” [All Fields] OR “randomized” [All Fields])
2 October 2024	MEDLINE (via OVID)	“football”, “soccer”, “F-MARC”, “FIFA 11+”, “The 11+”, “warm-up programme”, “injury prevention programme”, “injury”, “injuries”, “RCT”, “clinical trial”	All	Language: English Year: 2006 onwards	1. (football OR soccer).mp. 2. (F-MARC OR FIFA 11+ OR The 11+ OR warm-up programme OR injury prevention programme).mp. 3. (injury OR injuries).mp. 4. (randomized controlled trial OR RCT OR clinical trial OR randomised).mp. 5. 1 AND 2 AND 3 AND 4
2 October 2024	Scopus	“football”, “soccer”, “F-MARC”, “FIFA 11+”, “The 11+”, “warm-up programme”, “injury prevention programme”, “injury”, “injuries”, “RCT”, “clinical trial”	Title, abstract, keywords	Language: English Year: 2006 onwards	TITLE-ABS-KEY((football OR soccer) AND (F-MARC OR FIFA 11+ OR The 11+ OR warm-up programme OR injury prevention programme) AND (injury OR injuries) AND (randomized OR randomised OR RCT OR clinical trial))
2 October 2024	SPORTDiscus	“football”, “soccer”, “F-MARC”, “FIFA 11+”, “The 11+”, “warm-up programme”, “injury prevention programme”, “injury”, “injuries”, “RCT”, “clinical trial”	All	Language: English Year: 2006 onwards	(football OR soccer) AND (F-MARC OR FIFA 11+ OR The 11+ OR warm-up programme OR injury prevention programme) AND (injury OR injuries) AND (randomized OR randomised OR RCT OR clinical trial)

**Table 4 muscles-04-00030-t004:** Characteristics of studies included in the systematic review, describing the methodological design, participants, intervention, frequency, duration, and outcome measures.

Study Details		Participants		Intervention				Outcome
Authors/Year of Publication	Design	Number of Subjects	Characteristics	Intervention	Frequency	Duration	Control	Measures
Soligard et al. [16]	Cluster RCT	1892	Intervention group (*n* = 1055) Age: 15.4 ± 0.7 Control group (*n* = 837) Age: 15.4 ± 0.7 Gender: Female Clubs from the 15–16-year age divisions organized by the Norwegian Football Association (Norway)	Players followed the FIFA 11+ programme, which included running, strength, plyometric, and balance exercises aimed at injury prevention, lasting about 20 min per session.	2–5 times per week	8 months	Players performed their usual warm-up routines.	Ankle injuries
Owoeye et al. [30]	Cluster RCT	416	Intervention group (*n* = 212) Age: 17.80 ± 0.94 Control group (*n* = 204) Age: 17.49 ± 1.10 Gender: Male Premier League Division of the Lagos Junior League (Nigeria)	Players followed the FIFA 11+ programme, which included running, strength, plyometric, and balance exercises aimed at injury prevention, lasting about 20 min per session.	2 times per week	6 months	Players continued with their usual non-structured warm-up during training, with no additional training programme.	Ankle injuries
Silvers-Granelli et al. [18]	RCT	1525	Intervention group (*n* = 675) Age: 20.40 ± 1.66 Control group (*n* = 850) Age: 20.68 ± 1.46 Gender: Male NCAA Division I and Division II (USA)	Players followed the FIFA 11+ programme, which included running, strength, plyometric, and balance exercises aimed at injury prevention, lasting about 20 min per session.	3 times per week	5 months	Followed typical warm-up routines, including running, static and dynamic stretching, direction changes, and short passing drills, lasting 5 to 45 min.	Ankle injuries

*n* = number of participants; RCT = randomised controlled trial.

**Table 5 muscles-04-00030-t005:** Injury rates per 1000 h of exposure in the intervention and control groups of the included studies with IRRs and 95% CIs.

Study	FIFA 11+				Control				IRR (95%CI)
	*n*	Ankle Injuries	Exposure Hours	Ankle Injuries/1000 h	*n*	Ankle Injuries	Exposure Hours	Ankle Injuries/1000 h	
Soligard et al. [16]	1055	51	49,899	1.022	837	52	45,428	1.145	0.89 (0.61–1.31)
Owoeye et al. [30]	212	10	51,017	0.196	204	30	61,045	0.491	0.40 (0.19–0.82)
Silvers-Granelli et al. [18]	675	59	35,226	1.675	850	115	44,212	2.601	0.65 (0.48–0.87)

*n* = number of participants; IRR = incidence rate ratio; CI = confidence interval.

**Table 6 muscles-04-00030-t006:** SoF table including total exposure hours, relative effects, anticipated absolute effects, difference, and GRADE certainty of evidence for ankle injury prevention outcomes.

Outcome	Total Exposure Hours (Studies)	Relative Effects (95% CI)	Anticipated Absolute Effects (95% CI)			Certainty of the Evidence (GRADE)	Comments
			Without FIFA 11+	With FIFA 11+	Difference		
Number of ankle injuries	286,827 (3 RCTs)	RR 0.67 (0.46 to 0.96)	1.31 per 1000 exposure hours (0.66 to 1.96 per 1000)	0.88 per 1000 exposure hours (0.60 to 1.25 per 1000)	0.43 fewer ankle injuries per 1000 exposure hours (0.06 fewer to 0.71)	⊕⊕⊕⊝ **Moderate** Downgraded for risk of bias.	Probably decreases the number of ankle injuries per 1000 exposure hours.

RCT = randomised controlled trial; RR = risk ratio; CI = confidence interval.

## Data Availability

No new data were created or analysed in this study. Data sharing is not applicable to this article.

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
