# Peer review of "The Impact of the FIFA 11+ Neuromuscular Training Programme on Ankle Injury Reduction in Football Players: A Systematic Review and Meta-Analysis"

_muscles, 2025, doi:10.3390/muscles4030030_

Round 1
Reviewer 1 Report
Comments and Suggestions for Authors
Reviewer Comments on Manuscript: “The Impact of the FIFA 11+ Injury Prevention Programme on Ankle Injury Reduction in Football Players: A Systematic Review”
This review tackles a relevant and practical question — whether FIFA 11+ can reduce ankle injuries — and does so with a solid methodological approach. It’s clearly written, sticks closely to PRISMA and GRADE standards, and the results (a 33% reduction in injury risk) are meaningful. It’s also refreshing to see ankle injuries examined separately rather than bundled under general injury incidence.
Strengths:
- Clear and well-defined inclusion criteria (PICO).
- Appropriate use of risk-of-bias tools.
- Addresses an important yet relatively overlooked area of football injury prevention.
Suggestions for Improvement:
- Small Study Pool:
Only three RCTs were included. This limits confidence in the pooled estimate and should be discussed more explicitly, especially in terms of publication bias and the potential benefit of including high-quality non-RCT data.
- Unexplored Heterogeneity:
The I² value (52%) suggests some variability that isn’t explained. Were there differences in age groups, competitive level, or programme adherence? A short discussion would help.
- Practical Implications:
Coaches and practitioners would benefit from knowing what level of programme adherence is necessary. Is 2 sessions/week sufficient? Are certain components more critical than others?
- Narrow Outcome Focus:
Injury incidence is useful, but severity and time-loss are just as important in practice. Including this in the discussion — perhaps citing Verhagen et al. (BJSM 2024) — would round out the picture.
Minor Points:
- There are some inconsistencies in the formatting of exposure hours.
- The forest plots mentioned were missing in the version reviewed — please ensure these are included.
Conclusion:
Overall, a well-conducted review that fills a useful gap in the literature. A few additions to the discussion would strengthen its relevance to both researchers and practitioners.
Recommendation: Minor to moderate revision.
Author Response
Comment 1:
Small Study Pool:
Only three RCTs were included. This limits confidence in the pooled estimate and should be discussed more explicitly, especially in terms of publication bias and the potential benefit of including high-quality non-RCT data.
Response 1:
We appreciate this important comment. We have now explicitly acknowledged the limitations posed by the small number of included RCTs, including the implications for statistical power and the inability to formally assess publication bias. Additionally, we have noted the potential value of including high-quality non-randomised studies in future reviews, particularly in elite football populations where RCTs are often impractical.
[Lines 383–393]
Comment 2:
Unexplored Heterogeneity:
The I² value (52%) suggests some variability that isn’t explained. Were there differences in age groups, competitive level, or programme adherence? A short discussion would help.
Response 2:
Thank you for highlighting this important point. We have now elaborated on possible sources of heterogeneity in the meta-analysis, including differences in participant demographics and intervention characteristics. This additional discussion clarifies how variability in age, sex, competitive level, session frequency, and programme duration may have contributed to the observed I² value.
[Lines 397–408]
Comment 3:
Practical Implications:
Coaches and practitioners would benefit from knowing what level of programme adherence is necessary. Is 2 sessions/week sufficient? Are certain components more critical than others?
Response 3:
Thank you for this insightful comment. We agree that providing clearer practical implications enhances the value of our findings. Therefore, we have added a detailed paragraph to the Discussion section addressing minimum implementation frequency, adherence insights from relevant studies, and key exercise components.
[Lines 418-427]
Comment 4:
Narrow Outcome Focus:
Injury incidence is useful, but severity and time-loss are just as important in practice. Including this in the discussion — perhaps citing Verhagen et al. (BJSM 2024) — would round out the picture.
Response 4:
Thank you for this valuable suggestion. We fully agree that focusing solely on injury incidence may limit the understanding of an intervention’s practical significance. In response, we have expanded the Discussion to reflect on injury severity and burden, incorporating the argument from Verhagen et al. [36], and emphasising the importance of including these measures in future research.
[Lines 453-462]
Comment 5:
There are some inconsistencies in the formatting of exposure hours.
Response 5:
Thank you for pointing this out. We have corrected formatting inconsistencies related to exposure hours and standardised the terminology throughout the manuscript to “injuries per 1,000 exposure hours.”
Comment 6:
The forest plots mentioned were missing in the version reviewed — please ensure these are included.
Response 6:
Thank you for this observation. The forest plots were included at the time of submission as Figures 3 and 4, both embedded in the submitted Word and PDF files and also provided separately in the ZIP file for figures. We have double-checked to ensure they are correctly formatted, labelled, and referenced in the revised manuscript.
[Page 11]
Reviewer 2 Report
Comments and Suggestions for Authors
General comments
Thank you for the opportunity in reviewing this manuscript, overall it is well written and designed. The results have clear conclusions and applications to practise. I do have 1 concern over the viability of only including 3 studies for a systematic review, is this appropriate?
I would also recommend that you consider looking into the compliance of the included studies, which may support your suggestion on the dose-response relationship.
I have some minor comments detailed below.
Specific comments
Abstract
L13 – A systematic review and meta-analysis is inanimate and cannot evaluate, please amend accordingly
L28 – “future research is needed” On what? Can you be more specific here.
L93 – Make “aim” plural to “aims”
Methods
L151 – Can you add the initials to clarify which authors did what?
L186 – define CIs on first use
L199 – How was the I2 statistic interpreted?
Results
L237 – 3 studies appropriate?
Discussion
L360 – “additional exposure”, would this be related to compliance?
L379 – 2 commas, needs amending
L387 – “dose-response”, would this be related to compliance?
L445-452 – I would delete this last paragraph, it seems out of place here. They are also very general, unlike the limitations highlighted above.
Author Response
Comment 1:
I do have 1 concern over the viability of only including 3 studies for a systematic review, is this appropriate?
Response 1:
Thank you for this important observation. We have added a paragraph to the Discussion explicitly addressing the limitations associated with the inclusion of only three RCTs. We acknowledged the reduced statistical power and limitations in generalisability, discussed the constraints in formally assessing publication bias with a small study pool, and proposed future consideration of high-quality non-randomised studies, particularly in elite football contexts.
[Lines 383–393]
Comment 2:
I would also recommend that you consider looking into the compliance of the included studies, which may support your suggestion on the dose-response relationship.
Response 2:
Thank you for this helpful recommendation. In response, we have added a paragraph to the Discussion that outlines practical recommendations for implementation frequency based on available data from the included studies. Although the studies did not directly examine a dose–response relationship with ankle injury outcomes, we highlight related findings on adherence and neuromuscular improvements.
[Lines 418-427]
Comment 3:
L13 – A systematic review and meta-analysis is inanimate and cannot evaluate, please amend accordingly
Response 3:
Thank you, we have revised the sentence for clarity.
[Line 13]
Comment 4:
L28 – “future research is needed” On what? Can you be more specific here.
Response 4:
We agree and have now specified the direction for future research.
[Lines 29-30]
Comment 5:
L93 – Make “aim” plural to “aims”
Response 5:
Corrected.
[Lines 96]
Comment 6:
L151 – Can you add the initials to clarify which authors did what?
Response 6:
We have clarified author contributions by adding initials.
[Lines 153-156]
Comment 7:
L186 – define CIs on first use
Response 7:
Thank you. We have now defined confidence intervals (CIs) at first mention.
[Line 190]
Comment 8:
L199 – How was the I2 statistic interpreted?
Response 8:
Thank you. We have now added an explanation of how I² values were interpreted based on Cochrane guidance.
[Lines 202-206]
Comment 9:
L237 – 3 studies appropriate?
Response 9:
We addressed this concern in the Discussion section by acknowledging the limitations of a small study pool and discussing implications for generalisability and publication bias.
[383-393]
Comment 10:
L360 – “additional exposure”, would this be related to compliance?
Response 10:
Thank you for pointing this out. We have clarified this sentence by explaining that “additional exposure” refers to greater adherence and cumulative neuromuscular training volume, which may influence injury reduction outcomes.
[Lines 365-371]
Comment 11:
Line 379:
Two commas — needs amending
Response 11:
Thank you for pointing this out. This issue has been resolved during the revision process, as the sentence was modified and the extra comma was removed.
[Line 397]
Comment 12:
L387 – “dose-response”, would this be related to compliance?
Response 12:
Thank you for this clarification request. We have added a paragraph to the Discussion to explicitly link the concept of dose–response to adherence. While none of the included studies stratified outcomes by adherence level, we now address this limitation and its implications for interpreting the intervention’s effectiveness.
[Lines 408-413]
Comment 13:
L445-452 – I would delete this last paragraph, it seems out of place here. They are also very general, unlike the limitations highlighted above.
Response 13:
Thank you for this observation. We have removed the final paragraph as suggested to maintain focus and avoid generality.
[Line 482]
Round 2
Reviewer 1 Report
Comments and Suggestions for Authors
General comments
I do not have any further concerns about this manuscript. The authors addressed all the issues I raised well enough.